# Analysis of *CACNA1C* and *KCNH2* Risk Variants on Cardiac Autonomic Function in Patients with Schizophrenia

**DOI:** 10.3390/genes13112132

**Published:** 2022-11-16

**Authors:** Alexander Refisch, Shoko Komatsuzaki, Martin Ungelenk, Andy Schumann, Ha-Yeun Chung, Susann S. Schilling, Wibke Jantzen, Sabine Schröder, Markus M. Nöthen, Thomas W. Mühleisen, Christian A. Hübner, Karl-Jürgen Bär

**Affiliations:** 1Department of Psychiatry and Psychotherapy, Jena University Hospital, 07743 Jena, Germany; 2Lab for Autonomic Neuroscience, Imaging and Cognition (LANIC), Department of Psychosomatic Medicine and Psychotherapy, Jena University Hospital, 07743 Jena, Germany; 3Institute of Human Genetics, Jena University Hospital, 07743 Jena, Germany; 4Section for Translational Neuroimmunology, Department of Neurology, Jena University Hospital, 07743 Jena, Germany; 5Institute of Human Genetics, University of Bonn, 53113 Bonn, Germany; 6Department of Genomics, Life and Brain Center, University of Bonn, 53113 Bonn, Germany; 7Institute of Neuroscience and Medicine (INM-1), Research Center Jülich, 52428 Jülich, Germany; 8Cécile and Oskar Vogt Institute of Brain Research, Medical Faculty, Heinrich Heine University Düsseldorf, 40225 Düsseldorf, Germany; 9Human Genomics Research Group, Department of Biomedicine, University of Basel, 4001 Basel, Switzerland

**Keywords:** schizophrenia, *CACNA1C*, *KCNH2*, cardiac autonomic dysfunction, heart rate variability, vagal function, QT variability, cardiac mortality

## Abstract

Background: Cardiac autonomic dysfunction (CADF) is a major contributor to increased cardiac mortality in schizophrenia patients. The aberrant function of voltage-gated ion channels, which are widely distributed in the brain and heart, may link schizophrenia and CADF. In search of channel-encoding genes that are associated with both CADF and schizophrenia, *CACNA1C* and *KCNH2* are promising candidates. In this study, we tested for associations between genetic findings in both genes and CADF parameters in schizophrenia patients whose heart functions were not influenced by psychopharmaceuticals. Methods: First, we searched the literature for single-nucleotide polymorphisms (SNPs) in *CACNA1C* and *KCNH2* that showed genome-wide significant association with schizophrenia. Subsequently, we looked for such robust associations with CADF traits at these loci. A total of 5 *CACNA1C* SNPs and 9 *KCNH2* SNPs were found and genotyped in 77 unmedicated schizophrenia patients and 144 healthy controls. Genotype-related impacts on heart rate (HR) dynamics and QT variability indices (QTvi) were analyzed separately in patients and healthy controls. Results: We observed significantly increased QTvi in unmedicated patients with CADF-associated risk in *CACNA1C* rs2283274 C and schizophrenia-associated risk in rs2239061 G compared to the non-risk allele in these patients. Moreover, unmedicated patients with previously identified schizophrenia risk alleles in *KCNH2* rs11763131 A, rs3807373 A, rs3800779 C, rs748693 G, and 1036145 T showed increased mean HR and QTvi as compared to non-risk alleles. Conclusions: We propose a potential pleiotropic role for common variation in *CACNA1C* and *KCNH2* associated with CADF in schizophrenia patients, independent of antipsychotic medication, that predisposes them to cardiac arrhythmias and premature death.

## 1. Introduction

Life expectancy in patients with schizophrenia is shortened by about 15–20 years [1,2]. Cardiovascular complications and an increased prevalence of sudden cardiac death (SCD) significantly contribute to premature deaths in schizophrenia patients [3,4]. In addition, it was suggested that 10% of “unexplained” deaths may be attributed to underlying cardiac arrhythmias and SCD [5]. Emerging evidence revealed an inherent genetic risk contributing to increased cardiac mortality in schizophrenia [6]. However, the underlying molecular mechanisms that link schizophrenia and impaired cardiac function remain elusive.

The autonomic nervous system plays a pivotal role in modulating cardiac electrophysiology and arrhythmogenesis [7]. Cardiac autonomic dysfunction (CADF) has been extensively described in acute and chronic schizophrenia patients, in unmedicated or drug-naïve patients with schizophrenia and their healthy first-degree relatives [8,9,10]. The main CADF features are: increased heart rates (HR), reduced HR variability/complexity, and increased variability of the QT interval, which is independent of premedication [11,12]. Associated genetic loci overlapping with both CADF and schizophrenia include genes encoding ion channel subtypes that are widely distributed in the brain and heart [13,14]. Since both neurons and cardiomyocytes are excitable cells, ion channel functions are crucial for their physiological activity [15].

Timothy syndrome (TS), which is caused by congenital or inherited mutations in a subunit of the L-type calcium channel Cav1.2, is a plausible example of how ion channel dysfunction can result in clinical conditions with both psychiatric and cardiac phenotypes [16]. L-type calcium currents play an important role in the excitation-contraction coupling of cardiomyocytes as well as in neuronal excitability [17,18]. Besides well-described pathogenic variants in *CACNA1C* leading to TS [19], long QT syndrome (LQT) [20], and Brugada syndrome (BrS) [21], single nucleotide polymorphisms (SNPs) within *CACNA1C* have been described as one of the most replicable and consistent associations in psychiatric genetics [22,23]. Moreover, rare variants in *CACNA1C* have also been implicated in schizophrenia [24]. Previous studies indicate that compounds, which affect calcium channels, could have a positive effect on treating schizophrenia [25,26].

The α subunit of a potassium ion channel Kv11.1 encoded by the hERG gene (*KCNH2*) could be involved in the molecular biological mechanism of inherent cardiac vulnerability in patients with schizophrenia for several reasons. *KCNH2* is best known for its contribution to the repolarisation of the heart. Thus, blocking hERG (*KCNH2*) channels results in cardiac arrhythmia, and a variety of *KCNH2* mutations can cause congenital long QT syndrome. In addition, *KCNH2* is of special interest for psychiatric research since Huffaker et al. identified a brain-specific isoform (*KCNH2-3.1*) with altered gating kinetics in schizophrenia patients, which may contribute to uncoordinated neuronal firing patterns [27]. Moreover, *KCNH2*-SNPs have been associated with schizophrenia, lower IQ, and lower cognitive processing speed [27,28].

Given the possible association of *CACNA1C* and *KCNH2* with CADF in schizophrenia, we searched the literature for significant associations with cardiac autonomic traits, QT prolongation, and/or schizophrenia. We hypothesized that these SNPs might be associated with CADF features in drug-free patients with schizophrenia.

## 2. Participants and Methods

### 2.1. Participants

In this study, 144 Caucasian healthy control subjects and 77 schizophrenia patients were enrolled.

Patients were recruited upon admission to the hospital or the outpatient department of Jena University Hospital when they were in the acute stage. Using the Structured Clinical Interview for DSM-IV (SCID), a diagnosis was made by a staff psychiatrist in accordance with DSM-IV criteria (Diagnostic and statistical manual of mental disorders, 4th edition, published by the American Psychiatric Association). The diagnosis was confirmed by an independent psychiatrist and was reevaluated after 3 months in case of first-episode psychosis. Only patients who had been off antipsychotic medication for at least 8 weeks prior to the trial were included. Control subjects were recruited from local residents.

All subjects were informed about the procedures one day in advance. Each participant signed an informed consent form approved by the Ethics Committee of the Jena University Hospital, Germany. Patients were informed that refusal to take part in the study does not affect any future medical care in our hospital. Every effort was made to ensure that patients were able to provide informed consent. Patients were only included after a psychiatrist had verified their ability to provide fully informed consent to the study protocol. To rule out any other somatic diseases, such as a history of hypertension, diabetes, or other cardiovascular diseases, all subjects underwent a screening program that included testing for drug residues, legal and illegal substances, a full clinical examination, a baseline echocardiogram (ECG), and standard laboratory parameters. The screening procedure was conducted by a staff psychiatrist prior to the autonomic assessment. Patients and controls who were on any medication (β-blockers, antiarrhythmics, tranquilizers, or antidepressants, for example) that affected the regulation of HR or blood pressure were not included. After the screening process, a staff psychiatrist used the Positive and Negative Syndrome Scale to quantify the severity of psychotic symptoms (PANSS) [29]. All subjects were instructed to abstain from smoking, heavy eating, and exercise two hours before the examination.

### 2.2. Assessment of Autonomic Function

Examinations were scheduled from 1 to 6 p.m. in a quiet room that was kept comfortably warm (22–24 °C). Subjects were instructed to remain as still as possible while breathing normally and relaxing. We recorded physiological signals using the MP150 system (BIOPAC Systems Inc., Goleta, CA, USA) for 30 min at a sampling frequency of 1000 Hz. Three electrodes were placed on the chest in the shape of a modified Einthoven triangle to record the ECG. Band-pass filters were applied to ECG signals between 0.05 and 35 Hz. Automatically detected RR-interval time series were carefully examined for ectopic beats or artifacts, which were then removed using linear interpolation [30].

Due to the high scanning frequency and, thus, temporal resolution of the ECG recordings, the obtained measures regarding RR intervals allow a reliable parameter calculation of heart rate variability (HRV) [31].We calculated typical HRV measurements in the frequency domain in accordance with the relevant standards [32]. After performing a Fast Fourier transformation on the RR-interval time series, we integrated spectral power in the low-frequency (LF; 0.04–0.15 Hz) and high-frequency bands (HF; 0.15–0.40 Hz). The LF/HF ratio has been proposed to describe sympatho-vagal balance, with high values indicating sympathetic dominance as the HF component is related to cardiovagal modulation and the LF component is linked to both sympathetic and parasympathetic influence [33,34]. LF/HF is widely applied to evaluate the sympatho-vagal balance, despite the fact that there is still disagreement regarding the precise interpretation of LF power [35,36,37,38]. In addition, the root-mean-square of successive differences (RMSSD) was calculated, which detects rapid fluctuations in heart rate and thus indicates the parasympathetic cardiac function [32].

Nonlinear complexity parameters defining regularities of HR time series were introduced to complement linear HRV parameters reflecting the variation of beat-to-beat intervals. The use of these innovative analyses increased the sensitivity for identifying autonomic dysfunction [39]. Compression entropy (Hc), which was introduced by Baumert and colleagues, is a frequently used parameter to describe nonlinear feature of HR time series [40]. Hc describes the extent to which HR time series can be compressed by searching for repeating patterns. The compression rate increases with the frequency of the repeating sequences, indicating a high regularity of the underlying time series.

In addition, *Bazett’s formula* (QTc) was used to adjust the mean QT interval for the heart rate (QTc). Furthermore, we used a QT variability algorithm which was introduced by Berger et al. [41]. In brief, a graphical interface of digitized ECG was applied. The operator provides the program with the beginning and the end of the QT wave template after using a peak detection algorithm to determine the time of the ‘R’ wave. By using the time-stretch model, this algorithm finds the QT interval for each beat. This algorithm produces beat-to-beat RR and QT intervals as its output. The sampling frequency for both signals was 4 Hz to ensure that the same length of time was used for the analysis. Prior to computing the spectral analysis, the RR and QT interval data were then detrended using the best-fit line. The instantaneous RR and QT time series recorded at 4 Hz were used to determine mean RR (mRR) and variance (RRv), as well as mean QT interval (mQT) and detrended QT interval variance (QTv). According to Berger et al., a normalized QT variability index (QTvi) was calculated using the formula: QTvi = Log10 ((QTv/mQT2)/(RRv/mRR2)) [41]. The QT interval and the RR variabilities (detrended), each adjusted for the appropriate mean, are represented by the log ratio in this index.

### 2.3. Marker Selection, Genetic Analyses, and Identification of Subgroups

We searched the literature for tagging SNPs in the two candidate genes *CACNA1C* and *KCNH2* that have been previously reported to be associated with CADF traits, long QT syndrome, and/or schizophrenia (Table 1). *CACNA1C* rs1006737 G > A (*p* = 1.09 × 10^−16^, rs4765905 G > C (*p* = 1.08 × 10^−16^), rs2007044 A > G (*p* = 2.63 × 10^−17^) and rs2239063 A > C (*p* = 5.39 × 10^−9^) have been strongly linked with schizophrenia [22,42]. Moreover, *CACNA1C* rs2283274 G > C has been reported to be associated with resting HR (*p* = 7.21× 10^−20^) [43,44]. Huffaker et al. identified *KCNH2* rs11763131 G > A, rs3807373 G > A, rs3807374 T > G, rs380779 T > C, rs748693 A > G, and rs1036145 C > T as risk variants for schizophrenia that predict lower intelligence quotient scores and speed of cognitive processing, altered memory-linked functional magnetic resonance imaging signals and increased *KCNH2-3.1* mRNA levels in postmortem hippocampus [27]. In addition, *KCNH2* rs2968864 T > C, rs4725982 C > T, and rs2072413 C > T have been reported to influence QT interval duration [45,46]. *KCNH2* rs4725982 has also been related to sudden cardiac death [47]. *KCNH2* rs1805120 G > A predisposes to acquired atrial fibrillation [48].

We created an AmpliSeq^TM^ Custom DNA Panel for Illumina^®^ (Illumina Inc., San Diego, CA, USA) to perform high-throughput sequencing, which contains the selected SNPs in *CACNA1C* and *KCNH2*.

We collected venous blood from the crook of the arm after the subjects read the informed consent form and signed it. The vessel was compressed above the sampling site, and one EDTA tube of blood was taken. It was first centrifuged and then frozen until analysis. Leukocytes from the peripheral blood were used to extract DNA using the QIAamp DNA Blood Mini and Maxi kits (Qiagen, Hilden, Germany). Following the manufacturer’s instructions, next-generation sequencing (NGS) was carried out on 10 ng of high-quality DNA from each participant using the AmpliSeqTM for Illumina^®^ methodology. In brief, endonucleases were used to break up participant DNA, which was then hybridized to biotinylated gene-specific probes that included Illumina paired-end sequencing motifs and indexing primers. Magnetic beads were used to trap hybridized molecules, which were then amplified by PCR and sequenced using the MiSeq technology (Illumina Inc., San Diego, CA, USA).

According to the identified risk status for the selected SNPs in *CACNA1C* and *KCNH2*, diagnostic groups (unmedicated patients with schizophrenia and healthy controls) were separately divided into two genotype subgroups. Genotypes that contain corresponding risk alleles were defined as risk genotype. Homozygote non-risk allele carriers were defined as non-risk genotype.

### 2.4. Statistical Analysis

SPSS for Windows (version 23.0, IBM, Jena, Germany) was used for statistical analyses.

By using the chi-square test implemented in the FINETTI tool (http://ihg.gsf.de/cgi-bin/hw/hwa1.pl, accessed on 4 February 2022 ), analyses of the Hardy–Weinberg Equilibrium were carried out separately for patients and controls.

MANOVAs and follow-up univariate ANOVAs were performed separately in unmedicated patients with schizophrenia and healthy controls to compare mHR, RMSSD, LF/HF, Hc, QTc, and QTvi between allelic risk in *CACNA1C* rs1006737 (GG vs. AG/AA), rs4765905 (GG vs. CG/CC), rs2007044 (AA vs. AG/GG), rs2239063 (AA vs. AC/CC) and rs2283274 (GG vs. GC/CC) as well as in *KCNH2* rs11763131 (GG vs. AG/AA), rs3807373 (GG vs. AG/AA), rs3807374 (TT vs. TG/GG), rs380779 (TT vs. TC/CC), rs748693 (AA vs. AG/GG), rs1036145 (CC vs. CT/TT), rs2968864 (TT vs. CT/CC) T > C, rs4725982 (CC vs. CT/TT), and rs1805120 (GG vs. AG/AA).

## 3. Results

Sociodemographic data are shown in Table 2.

There is a trend for healthy controls to be younger and to smoke less in pair-wise comparisons between diagnostic groups. In addition, there are significant differences in daily coffee intake between patients with schizophrenia and healthy controls.

### Associations of CACNA1C and KCNH2 Risk Variants with Parameters of Cardiac Autonomic Dysfunction

The distribution of allele frequencies is presented in Table 3 for *CACNA1C* SNPs and Table 4 for *KCNH2* SNPs. Genotype distributions of the selected SNPs were all in Hardy–Weinberg Equilibrium in each diagnostic group. Pair-wise analysis showed that *KCNH2* rs11763131, rs3807373, rs3807374, rs380779, rs748693, rs1036145, and rs2072413 are in strong linkage disequilibrium (LD) in Europeans (r^2^ > 0.8, CEU from 1000 Genomes Project) [51]. Moreover, *KCNH2* rs1805120 is correlated with rs4725982. Among the *CACNA1C* variants, rs1006737 is correlated with rs2007044 and rs4765905. Since both target genes are located on different chromosomes, *CACNA1C* and *KCNH2* SNPs assort independently and are unlinked.

Next, the interaction effect of *genotype risk* × *diagnosis* on cardiac autonomic parameters was tested. We observed significant interaction effects (*p* < 0.001) for all selected SNPs on the main variables (mHR, RMSSD, LF/HF, CE, QTc, QTvi).

Regarding age, BMI, smoking behavior, athletic activities, coffee consumption, and psychometric scales, there were no significant differences in all investigated SNPs.

We further tested whether genotypes with identified risk alleles in *CACNA1C* affect cardiac autonomic parameters in diagnostic groups. MANOVAs showed significant main effects for *CACNA1C* rs2283274 (GG vs. CG/CC) (F(6,70) = 3.012, *p* = 0.011) and rs2239061 (AA vs. AG/GG) (F(6,70) = 2.235, *p* = 0.050) in schizophrenia patients, but not in healthy controls. Follow-up ANOVAs showed significant group differences in QTc (F(1,75) = 4.277, *p* = 0.042) and QTvi (F(1,75) = 9.562, *p* = 0.003) comparing genotype risk in rs2283274 as well as in QTvi (F(1,75) = 4.515, *p* = 0.037) comparing genotype risk in rs2239061 G > C (Table 3, Figure 1).

Next, we performed another set of MANOVAs testing the main effect of allelic risk in *KCNH2* SNPs on cardiac autonomic parameters. In schizophrenia patients we found a significant main effect for rs11763131 (GG vs. AG/AA) (F(6,70) = 3.067, *p* = 0.010), rs3807373 (GG vs. AG/AA) (F(6,70) = 2.625, *p* = 0.024), rs3800779 (TT vs. CT/CC) (F(6,70) = 2.773, *p* = 0.018), rs748693 (AA vs. AG/GG) (F(6,70) = 2.589, *p* = 0.025) and rs1036145 (CC vs. CT/TT) (F(6,70) = 2.312, *p* = 0.043). No differences were found in healthy controls comparing cardiac autonomic parameters between *KCNH2* risk alleles. Follow-up ANOVAs comparing cardiac autonomic parameters between *KCNH2* risk alleles in schizophrenia patients are listed in Table 3 and exemplarily displayed for rs3800779 T > C in Figure 2.

Finally, we tested for the main effect of genotypes (e.g., AA vs. AG vs. GG) of these SNPs on cardiac autonomic parameters. MANOVAs showed significant main effects for *CACNA1C* rs2283274 (GG vs. CG vs. CC) (F(12,138) = 1.865, *p* = 0.045) and rs2239063 (AA vs. AC vs. CC) (F(12,138) = 1.900, *p* = 0.039) in unmedicated patients with schizophrenia, but not in controls. Follow-up ANOVAs revealed no differences between single genotypes in *CACNA1C* rs2283274 G > C and rs2239063 A > C in any cardiac autonomic parameter.

Comparing genotypes in *KCNH2* SNPs, MANOVAs showed significant main effects on cardiac autonomic parameters in schizophrenia patients for rs11763161 (GG vs. AG vs. AA) (F(12,138) = 1.862, *p* = 0.044), rs748693 (AA vs. AG vs. GG) (F(12,138) = 2.003, *p* = 0.028) and rs1036145 (CC vs. CT vs. TT) (F(12,138) = 1.860, *p* = 0.044). Follow-up ANOVAs revealed significant main effects for rs11763131 (GG vs. AG vs. AA) on QTvi (F(2,74) = 3.711, *p* = 0.029). Post-hoc-t-tests demonstrated no significant differences in QTvi between non-risk homozygous rs11763131 G, heterozygous and homozygous rs11763131 A. For *KCNH2* rs748693 (AA vs. AG vs. GG) follow-up, ANOVAs showed significant main effects on mHR (F(2,74) = 4.373, *p* = 0.016), QTc (F(2,74) = 4.029, *p* = 0.022) and QTvi (F(2,74) = 3.705, *p* = 0.029). Pair-wise comparisons demonstrated significant differences in mHR between rs748693 non-risk allele homozygotes and risk allele homozygotes (AA vs. GG; *p* = 0.013), in QTc between heterozygotes and risk allele homozygotes (AG vs. GG; *p* = 0.018) and in QTvi between non-risk allele homozygotes and risk allele homozygotes (AA vs. GG; *p* = 0.025). Comparing genotypes in *KCNH2* rs1036145 (CC vs. CT vs. TT), follow-up ANOVAs revealed significant main effects on mHR (F(2,74) = 3.850, *p* = 0.026) and QTc (F(2,74) = 3.397, *p* = 0.039). In post-hoc-t-test mHR (*p* = 0.021) was significantly increased in rs1036145 risk allele homozygotes (TT) compared to non-risk allele homozygotes (CC). Moreover, rs1036145 risk allele homozygotes (TT) demonstrated significantly increased QTc (*p* = 0.33) compared to rs1036145 heterozygotes (Arking et al.).

## 4. Discussion

In the present study, we provide first support for the impact of the two candidate susceptibility genes, *CACNA1C* and *KCNH2,* on CADF in schizophrenia. This includes two major findings: Unmedicated schizophrenia patients with previously identified genetic risk status for schizophrenia in *CACNA1C* rs2239063 A > C and five *KCNH2* SNPs in strong LD demonstrate significant altered HR dynamics and QTvi compared to non-risk genotypes in these patients. Second, we observed significant cardiac autonomic impairments in unmedicated schizophrenia patients carrying the *CACAN1C* rs2238274-C allele, which achieved genome-wide significance for resting HR [43,44].

The human *CACNA1C* gene codes for the pore-forming Ca_V_1.2 subunit protein of the cardiac L-type voltage-gated calcium channels contribute prominently to normal cardiac repolarization [52] and neurological functions, including synaptic plasticity, neuronal survival, memory formation and learning [53]. The *CACNA1C* locus has been linked to schizophrenia, which is one of the most conclusive findings from genetic research on mental health [54]. Patients carrying the *CACNA1C* rs2283274-C allele showed significant QT prolongations, which are best known to predispose to life-threatening cardiac arrhythmias such as Torsade de Pointes (Moss, 1999). Moreover, QTvi is significantly increased in patients carrying *CACNA1C* rs2284274 C and rs2239063 C allele. QTvi is a normalized measure of beat-to-beat QT variability to heart rate HRV [41] and provides information on the phase in which the heart is most susceptible to arrhythmias [55]. Notably, both increased QTvi and prolonged QT intervals have been reported in patients with schizophrenia compared to healthy controls, even in the absence of antipsychotic medication [12,56], suggesting repolarization lability to be a certain biological feature of schizophrenia. Regarding this, we report a potential role for *CACNA1C* in this feasible physiological mechanism linking cardiovascular disease and schizophrenia.

The precise functional consequences of the *CACNA1C* SNPs rs2283274 and rs2239063 are yet to be clarified. *CACNA1C* rs2283274 C achieved genome-wide significance (*p* < 5 × 10^−8^) in two independent GWAS (all European descent) for resting HR (Eppinga et al., 2016; Ramírez et al., 2018). *CACNA1C* rs2239063 C was associated with schizophrenia (*p* = 1.93 × 10^−8^) and treatment response to olanzapine (*p* = 1 × 10^−8^) at a genome-wide significance level [22,57]. Both SNPs, *CACNA1C* rs2283274 and rs2239063, are located in noncoding sequences that are enriched for expression quantitative trait loci, suggesting that dysregulation of transcriptional control might be mechanistically relevant.

All of the five *KCNH2*-SNPs with an impact on CADF parameters in our patient cohort have been previously reported by Huffaker et al. to be associated with schizophrenia and with the expression of a truncated *KCNH2* transcript, referred to as *KCNH2-3.1*, in the postmortem hippocampus of both schizophrenia patients and healthy subjects. Moreover, risk-associated alleles have been previously demonstrated to predict schizophrenia-relevant phenotypes such as lower IQ scores, lower cognitive processing speed, decreased hippocampus gray matter volume, and altered memory-linked fMRI signal [27,58,59].Since these markers are in moderate to strong linkage disequilibrium, the region of interest maps to a ∼3 kb segment of intron 2 of *KCNH2* as a potential susceptibility locus.

Here, we demonstrate significantly increased HR and QTvi in unmedicated patients carrying schizophrenia-associated alleles in the *KCNH2*-SNPs reported by Huffaker et al. [27]. An increase in QTvi, which is caused either by increased QT variability or reduced HRV, has been shown to be associated with increased sympathetic activity [60,61]. Moreover, although not significant, we observed a trend towards increased LF/HF ratios and decreased RMSSD in patients with schizophrenia-associated risk status in these *KCHN2* SNPs, indicating a shift in sympatho-vagal balance to the disadvantage of vagal modulation. In sum, our findings suggest that schizophrenia risk in *KCNH2* may predispose schizophrenia patients to a well-described loss of vagal function in schizophrenia which is probably accompanied by increased sympathetic activity. This constellation is known to be associated with the development of cardiac disease, including coronary heart disease, and has been identified as an independent risk factor for premature death in various diseases [62,63,64].

Well-described genetic variants in *KCNH2* have been found to either cause the congenital form or predispose for the acquired form of long QT and short QT syndromes [65,66]. As QT-prolongation effects of antipsychotic medication result from blockade of the hERG (*KCNH2*)-channel in the heart [58,67], the hERG-channel has been considered an “antitarget” in the treatment of schizophrenia [68]. Intriguingly, Heide et al. demonstrated that risperidone caused a stronger in vitro blockade of the alternatively spliced *KCNH2-3.1* isoform than the full-length *KCHN2-1A* channel, which was associated with a better treatment response in patients with a genetic risk profile of overexpressing *KCNH2-3.1* [68,69]. Notably, *KCNH2-3.1* abundance is similar in the brain when compared with the full-length isoform A1 but 1000-fold lower in the heart. Thus, the observed genetic associations with CADF features are most likely not due to the alternatively spliced isoform *KCNH2-3.1* at the cardiac level [28,70]. Nevertheless, risk alleles could also be coupled with another SNP, which was not genotyped, or may affect gene expression in the heart. Taken together, these data hold the promise of optimizing response to antipsychotic medication by including genotype data on the one hand [68] and of improving drug safety by giving information on underlying individual cardiac vulnerability on the other.

By providing evidence for other genetic factors that might be associated with CADF features in drug-free patients with schizophrenia [71,72], the present work stipulates the assumption that CADF is an endophenotype of schizophrenia. Endophenotypes reducing the phenotypic complexity of mental disorders hold promise to shed light on the biological mechanisms underlying schizophrenia [73]. In this regard, CADF, which is associated with multiple aspects of schizophrenia pathology and the development of cardiometabolic comorbidities, may have the potential to uncover genetic connections between cardiac and neural phenotypes in schizophrenia [74]. Future studies following a similar approach to ours may help to further consolidate a potential pathophysiological link between schizophrenia and cardiovascular disease.

Our study has some limitations. First, we were unable to match patients and healthy controls in terms of all variables that might have an impact on cardiac autonomic function in order to achieve the maximum sample size. As a result, there are significant differences between patients and controls in influencing variables such as coffee consumption, smoking habits, and physical activity. Secondly, since it is effortful to perform an elaborated analysis of cardiac autonomic function in drug-free patients with schizophrenia, our sample size is still limited. Hence, we suggest a potential association between *CACNA1C* and *KCNH2* variants and CADF in schizophrenia, which needs to be replicated in larger cohorts with higher statistical power.

## 5. Conclusions

Here, we report novel candidate genetic associations with CADF in unmedicated patients with schizophrenia, which might contribute to increased cardiac mortality in these patients. Our results bolster the notion of considering CADF as an endophenotype of the disease. Genetic risk in *CACNA1C*, as one of the most established susceptibility genes for schizophrenia, may also have an impact on cardiac autonomic function in schizophrenia patients. Moreover, schizophrenia patients carrying risk alleles that have been previously associated with higher mRNA levels of schizophrenia-related, primate-specific, brain isoform *KCNH2-3.1* demonstrate significant cardiac autonomic impairments. Further studies are needed to investigate the potential pleiotropic role of *CACNA1C* and *KCNH2* variation in the disease architecture of schizophrenia and CADF.

## Figures and Tables

**Figure 1 genes-13-02132-f001:**
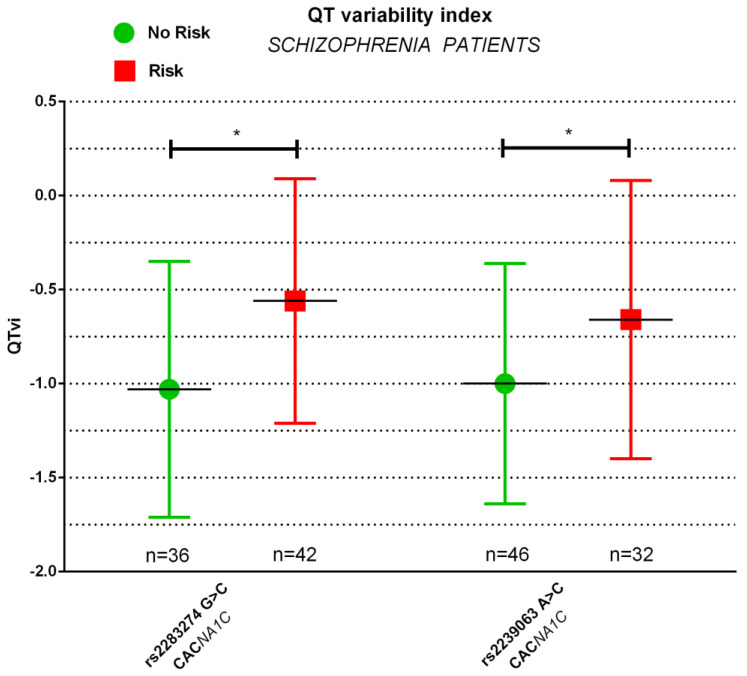
Error bar chart illustrating differences in QTvi (bpm) between genotype risk in *CACNA1C* rs2283274 (GG vs. CG/CC) and rs2239063 (AA vs. AC/CC) in unmedicated patients with schizophrenia, * *p* < 0.05, *p*-value resulting from ANOVAs. Error bar chart illustrating differences in QTvi (bpm) between genotype risk in CACNA1C rs2283274 (GG vs. CG/CC) and rs2239063 (AA vs. AC/CC) in unmedicated patients with schizophrenia, * *p* < 0.05, *p*-value resulting from ANOVAs.

**Figure 2 genes-13-02132-f002:**
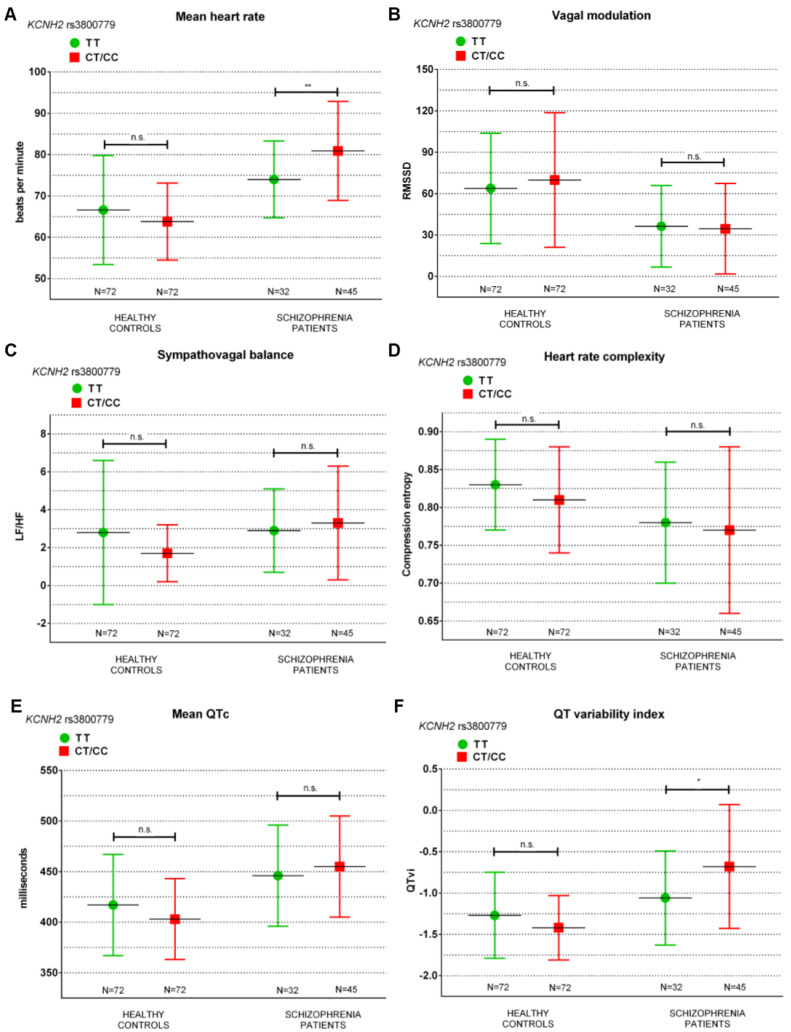
(**A**–**F**) Error bar chart illustrating differences in mHR (bpm) (**A**), RMSSD (**B**), LF/HF (**C**), Hc (**D**), QTc (**E**), and QTvi (**F**) between genotype risk in *KCNH2* rs3800779 (TT vs. CT/CC) separately in unmedicated schizophrenia patients and healthy controls, * *p* < 0.05, ** *p* < 0.01, *p*-value resulting from ANOVAs. Abbrev.: Not significant (n.s.).

**Table 1 genes-13-02132-t001:** Tagging SNPs in the two candidate genes CACNA1C and KCNH2 that have been previously reported to be associated with CADF traits, long QT syndrome, and/or schizophrenia.

SNP	HGVS Nomenclature	Annotation	Trait	Allele E/O	*p* Value	Source
rs11763131	NC_000007.14:g.150971094G>A	upstream	schizophrenia	A/G	0.07	[27]
rs3807373	NC_000007.14:g.150971633G>A	upstream	schizophrenia	A/G	0.013	
rs3807374	NC_000007.14:g.150971258T>C	upstream	schizophrenia	G/T	0.017	
rs3800779	NC_000007.14:g.150974126C>A	upstream	schizophrenia	C/T	0.0054	
rs748693	NC_000007.14:g.150974349A>G	upstream	schizophrenia	G/A	0.027	
rs1036145	NC_000007.14:g.150977342C>G	upstream	schizophrenia	T/C	0.015	
rs2968864	NC_000007.14:g.150925074T>C	downstream	QT interval	C/T	8 × 10^−16^	[46]
rs4725982	NC_000007.14:g.150940775C>G	downstream	QT interval sudden cardiac death	T/C	5 × 10^−16^	[46,47]
rs2072413	NC_000007.14:g.150950881C>T	intronic	QT interval	T/C	1 × 10^−49^	[45]
rs1805120	NC_000007.14:g.150952443G>A NM_000238.4:c.1539C>T NP_000229.1:p.Phe513=	synonymous variant	acquired atrial fibrillation	A/G	0.021	[48]
SNP	HGVS nomenclature	Annotation	Trait	Allele E/O	*p* value	Source
rs1006737	NC_000012.12:g.2236129G>A	intronic	schizophrenia	A/G	1.09 × 10^−16^	[49]
rs4765905	NC_000012.12:g.2240418G>A	intronic	schizophrenia	C/G	1.08 × 10^−16^	Psychiatric Genomics Consortium (PGC) [50]
rs2007044	NC_000012.12:g.2235794A>G	intronic	schizophrenia	G/A	2.63 × 10^−17^	[22,42]
rs2239063	NC_000012.12:g.2402665A>C	intronic	schizophrenia	C/A	5.39 × 10^−9^	PGC [50]
rs2283274	NC_000012.12:g.2075300G>C	intronic	resting heart rate	C/G	7.21 × 10^−20^	[43,44]

Abbrev.: Effect allele/other alleles (E/O).

**Table 2 genes-13-02132-t002:** Demographic Characteristics of Healthy Controls and Unmedicated Schizophrenia Patients.

	Diagnostic Group		*p*
	Healthy Controls	Schizophrenia Patients	
Soziodemographic Data			
N	144	77	NA
age (y)	29.9 ± 6.9 (19 to 44 y)	33.1 ± 11.5 (19 to 49 y)	0.009
gender (f/m)	73/71	45/32	n.s.
smoker status (y/n)	30/114	44/33	<0.001
cig. per day	1.3 ± 3.3	7.3 ± 9.5	<0.001
cups of coffee a day	1.4 ± 1.4	2.4 ± 2.3	0.025
BMI (m/kg^2^)	23.1 ± 3.4	23.0 ± 8.0	n.s.
hours of sport per week	1.9 ± 2.0	0.9 ± 1.4	n.s.
Psychopathology			
PANSS gen	NA	41.9 ± 11.4	NA
PANSS pos	NA	22.2 ± 6.3	NA

Data expressed as mean (SD). *p*-values resulting from ANOVAs. Abbrev.: Sample size (N), Not applicable (NA), Not significant (n.s.), Body mass index (BMI), PANSS: Positive and Negative Syndrome Scale [29].

**Table 3 genes-13-02132-t003:** Main Effect of *CACNA1C* Genotype Risk on Cardiac Autonomic Parameters.

	*CACNA1C* rs2283274 G > C					
	Healthy Controls			Schizophrenia Patients		
MAF	C = 0.16 (45/288)			C = 0.22 (34/154)		
GG	0.71 (103/144)			0.58 (45/77)		
CG	0.26 (37/144)			0.39 (30/77)		
CC	0.03 (4/144)			0.03 (2/77)		
χ^2^	0.0011			0.014		
CAF	GG	CG/CC	*p*	GG	CG/CC	*p*
N	103	41	n.s.	45	32	0.011
mHR	65.4 ± 11.8	64.5 ± 10.6	n.s.	77.5 ± 12.5	78.7 ± 9.8	n.s.
LF/HF	2.1 ± 2.1	2.6 ± 4.4	n.s.	3.5 ± 2.9	2.6 ± 2.3	n.s.
RMSSD	66.5 ± 41.2	67.7 ± 52.4	n.s.	34.0 ± 30.9	37.1 ± 32.3	n.s.
Hc	0.83 ± 0.06	0.81 ± 0.07	n.s.	0.77 ± 0.11	0.78 ± 0.09	n.s.
QTc	0.413 ± 0.04	0.404 ± 0.04	n.s.	0.441 ± 0.05	0.466 ± 0.05	0.042
QTvi	−1.35 ± 0.46	−1.33 ± 0.46	n.s.	−1.03 ± 0.68	−0.56 ± 0.65	0.003
	*CACNA1C* rs2007044 A > G					
	Healthy Controls			Schizophrenia Patients		
MAF	G = 0.38 (110/288)			G = 0.34 (52/154)		
AA	0.40 (58/144)			0.45 (35/77)		
AG	0.43 (62/144)			0.42 (32/77)		
GG	0.17 (24/144)			0.13 (10/77)		
χ^2^	0.0085			0.0041		
CAF	AA	AG/GG	*p*	AA	AG/GG	*p*
N	58	86	n.s.	35	42	n.s.
mHR	65.9 ± 13.7	64.7 ± 9.7	n.s.	77.8 ± 10.8	78.2 ± 11.9	n.s.
LF/HF	2.8 ± 4.2	1.9 ± 1.6	n.s.	3.2 ± 2.9	3.1 ± 2.5	n.s.
RMSSD	61.7 ± 38.5	70.3 ± 48.0	n.s.	32.0 ± 30.5	38.0 ± 32.1	n.s.
Hc	0.82 ± 0.07	0.82 ± 0.06	n.s.	0.76 ± 0.12	0.79 ± 0.09	n.s.
QTc	0.412 ± 0.05	0.409 ± 0.04	n.s.	0.449 ± 0.05	0.453 ± 0.05	n.s.
QTvi	−1.37 ± 0.45	−1.33 ± 0.47	n.s.	−0.86 ± 0.57	−0.81 ± 0.80	n.s.
	*CACNA1C* rs1006737 G > A					
	Healthy Controls			Schizophrenia Patients		
MAF	A = 0.34 (99/288)			A = 0.29 (44/154)		
GG	0.46 (66/144)			0.53 (41/77)		
AG	0.40 (57/144)			0.36 (28/77)		
AA	0.14 (21/144)			0.10 (8/77)		
χ^2^	0.0118			0.0106		
CAF	GG	AG/AA	*p*	GG	AG/AA	*p*
N	66	78	n.s.	41	36	n.s.
mHR	65.2 ± 13.2	65.2 ± 9.8	n.s.	77.8 ± 11.7	78.2 ± 11.2	n.s.
LF/HF	2.8 ± 4.0	1.8 ± 1.6	n.s.	3.0 ± 2.8	3.2 ± 2.6	n.s.
RMSSD	61.9 ± 37.5	71.0 ± 49.5	n.s.	35.3 ± 36.3	35.3 ± 25.0	n.s.
Hc	0.82 ± 0.07	0.82 ± 0.06	n.s.	0.76 ± 0.11	0.79 ± 0.08	n.s.
QTc	0.408 ± 0.05	0.412 ± 0.04	n.s.	0.451 ± 0.05	0.452 ± 0.05	n.s.
QTvi	−1.38 ± 0.44	−1.31 ± 0.48	n.s.	−0.87 ± 0.61	−0.80 ± 0.81	n.s.
	*CACNA1C* rs4765905 G > C					
	Healthy Controls			Schizophrenia Patients		
MAF	C = 0.33 (96/288)			C = 0.29 (44/154)		
GG	0.47 (68/144)			0.53 (41/77)		
CG	0.39 (56/144)			0.36 (28/77)		
CC	0.14 (20/144)			0.11 (8/77)		
χ^2^	0.0155			0.0158		
CAF	GG	CG/CC	*p*	GG	CG/CC	*p*
N	68	76	n.s.	41	36	n.s.
mHR	65.1 ± 13.1	65.3 ± 9.9	n.s.	77.8 ± 11.7	78.2 ± 11.2	n.s.
LF/HF	2.8 ± 3.9	1.7 ± 1.5	n.s.	3.0 ± 2.8	3.2 ± 2.6	n.s.
RMSSD	61.1 ± 37.3	72.0 ± 49.7	n.s.	35.3 ± 36.3	35.3 ± 25.0	n.s.
Hc	0.82 ± 0.07	0.82 ± 0.06	n.s.	0.76 ± 0.11	0.79 ± 0.08	n.s.
QTc	0.408 ± 0.05	0.412 ± 0.04	n.s.	0.451 ± 0.05	0.452 ± 0.05	n.s.
QTvi	−1.38 ± 0.43	−1.31 ± 0.48	n.s.	−0.87 ± 0.61	−0.80 ± 0.81	n.s.
	*CACNA1C* rs2239063 A > C					
	Healthy Controls			Schizophrenia Patients		
MAF	C = 0.34 (99/288)			C = 0.29 (44/154)		
AA	0.44 (64/144)			0.52 (40/77)		
AC	0.42 (61/144)			0.39 (30/77)		
CC	0.14 (19/144)			0.09 (7/77)		
χ^2^	0.0059			0.0019		
CAF	AA	AC/CC	*p*	AA	AC/CC	*p*
N	64	80	n.s.	40	37	0.050
mHR	65.0 ± 9.3	65.3 ± 13.0	n.s.	79.0 ± 11.1	77.0 ± 11.8	n.s.
LF/HF	2.2 ± 2.4	2.3 ± 3.4	n.s.	3.6 ± 3.1	2.6 ± 2.1	n.s.
RMSSD	67.3 ± 41.7	66.4 ± 46.8	n.s.	37.1 ± 36.2	33.3 ± 25.4	n.s.
Hc	0.82 ± 0.07	0.82 ± 0.06	n.s.	0.79 ± 0.08	0.76 ± 0.12	n.s.
QTc	0.411 ± 0.04	0.409 ± 0.05	n.s.	0.447 ± 0.04	0.456 ± 0.06	n.s.
QTvi	−1.35 ± 0.48	−1.34 ± 0.45	n.s.	−1.00 ± 0.64	−0.66 ± 0.74	0.037

Distribution of allele frequencies. Other data expressed as mean (SD). *p*-values resulting from MANOVAs and follow-up ANOVAs. χ^2^ indicates the results from the Hardy–Weinberg equilibrium test. Abbrev.: Not significant (n.s.), Minor Allele Frequency (MAF), Mean Heart Rate (mHR), Root Mean Sum of Squared Distance (RMSSD), Heart Rate Low-Frequency/High-Frequency ratio (LF/HF), Compression entropy (Hc), Mean QT interval corrected for heart rate (QTc), QT variability index (QTvi).

**Table 4 genes-13-02132-t004:** Main Effect of *KCNH2* Genotype Risk on Cardiac Autonomic Parameters.

	*KCNH2* rs2968864 T > C					
	Healthy Controls			Schizophrenia Patients		
MAF	C = 0.28 (80/288)			C = 0.27 (41/154)		
TT	0.48 (69/144)			0.51 (39/77)		
CT	0.47 (68/144)			0.45 (35/77)		
CC	0.05 (6/144)			0.04 (3/77)		
χ^2^	0.0235			0.0241		
CAF	TT	CT/CC	*p*	TT	CT/CC	*p*
N	70	74	n.s.	39	38	n.s.
mHR	64.1 ± 9.4	66.1 ± 13.1	n.s.	78.9 ± 11.6	77.2 ± 11.3	n.s.
LF/HF	1.8 ± 1.4	2.7 ± 3.8	n.s.	3.4 ± 3.1	2.8 ± 2.1	n.s.
RMSSD	66.5 ± 41.8	67.1 ± 47.2	n.s.	35.5 ± 32.7	35.1 ± 30.2	n.s.
Hc	0.82 ± 0.06	0.82 ± 0.07	n.s.	0.77 ± 0.12	0.78 ± 0.07	n.s.
QTc	0.404 ± 0.04	0.416 ± 0.05	n.s.	0.463 ± 0.05	0.439 ± 0.05	n.s.
QTvi	−1.40 ± 0.42	−1.29 ± 0.49	n.s.	−0.67 ± 0.76	−1.01 ± 0.61	n.s.
	*KCNH2* rs4725982 C > T					
	Healthy Controls			Schizophrenia Patients		
MAF	T = 0.21 (60/288)			T = 0.18 (28/154)		
CC	0.63 (90/144)			0.68 (52/77)		
CT	0.33 (48/144)			0.29 (22/77)		
TT	0.04 (6/144)			0.03 (3/77)		
χ^2^	0.0002			0.0000		
CAF	CC	CT/TT	*p*	CC	CT/TT	*p*
N	90	54	n.s.	52	25	n.s.
mHR	66.0 ± 12.8	63.7 ± 8.7	n.s.	76.3 ± 11.1	81.6 ± 11.2	n.s.
LF/HF	2.3 ± 3.3	2.2 ± 2.2	n.s.	2.9 ± 2.5	3.6 ± 3.1	n.s.
RMSSD	67.9 ± 48.3	65.0 ± 37.6	n.s.	39.0 ± 35.1	27.5 ± 19.7	n.s.
Hc	0.82 ± 0.07	0.82 ± 0.07	n.s.	0.79 ± 0.09	0.75 ± 0.12	n.s.
QTc	0.413 ± 0.05	0.405 ± 0.04	n.s.	0.447 ± 0.05	0.461 ± 0.06	n.s.
QTvi	−1.32 ± 0.48	−1.38 ± 0.43	n.s.	−0.87 ± 0.72	−0.76 ± 0.68	n.s.
	*KCNH2* 1,805,120 G > A					
	Healthy Controls			Schizophrenia Patients		
MAF	A = 0.28 (80/288)			A = 0.26 (40/154)		
GG	0.54 (78/144)			0.52 (40/77)		
AG	0.36 (52/144)			0.44 (34/77)		
AA	0.10 (14/144)			0.04 (3/77)		
χ^2^	0.0115			0.0206		
CAF	GG	AG/AA	*p*	GG	AG/AA	*p*
N	78	66	n.s.	40	37	n.s.
mHR	64.6 ± 9.5	65.9 ± 13.3	n.s.	78.9 ± 11.6	77.0 ± 11.4	n.s.
LF/HF	2.0 ± 1.8	2.6 ± 4.0	n.s.	3.4 ± 3.1	2.9 ± 2.1	n.s.
RMSSD	66.2 ± 43.5	67.4 ± 46.1	n.s.	35.5 ± 32.7	32.9 ± 26.4	n.s.
Hc	0.82 ± 0.07	0.83 ± 0.06	n.s.	0.77 ± 0.12	0.78 ± 0.07	n.s.
QTc	0.405 ± 0.04	0.417 ± 0.05	n.s.	0.463 ± 0.05	0.439 ± 0.05	n.s.
QTvi	−1.36 ± 0.47	−1.31 ± 0.47	n.s.	−0.67 ± 0.76	−1.04 ± 0.61	n.s.
	*KCNH2* rs2072413 C > T					
	Healthy Controls			Schizophrenia Patients		
MAF	T = 0.31 (90/288)			T = 0.27 (41/154)		
CC	0.47 (67/144)			0.52 (40/77)		
CT	0.44 (64/144)			0.44 (33/77)		
TT	0.09 (13/144)			0.04 (4/77)		
χ^2^	0.0008			0.0206		
CAF	CC	CT/TT	*p*	CC	CT/TT	*p*
N	67	77	n.s.	40	37	n.s.
mHR	65.2 ± 9.6	65.1 ± 12.9	n.s.	78.5 ± 11.6	77.4 ± 11.3	n.s.
LF/HF	1.8 ± 1.4	2.6 ± 3.8	n.s.	3.3 ± 3.2	2.9 ± 2.1	n.s.
RMSSD	64.6 ± 42.2	68.8 ± 46.6	n.s.	38.0 ± 35.6	32.4 ± 26.1	n.s.
Hc	0.81 ± 0.07	0.83 ± 0.07	n.s.	0.77 ± 0.12	0.78 ± 0.07	n.s.
QTc	0.408 ± 0.04	0.412 ± 0.05	n.s.	0.459 ± 0.05	0.443 ± 0.05	n.s.
QTvi	−1.35 ± 0.46	−1.34 ± 0.46	n.s.	−0.72 ± 0.74	−0.96 ± 0.66	n.s.
	*KCNH2* rs11763131 G > A					
	Healthy Controls			Schizophrenia Patients		
MAF	A = 0.25 (71/288)			A = 0.34 (52/154)		
GG	0.58 (84/144)			0.44 (34/77)		
AG	0.34 (49/144)			0.44 (34/77)		
AA	0.08 (11/144)			0.12 (9/77)		
χ^2^	0.0087			0.0004		
CAF	GG	AG/AA	*p*	GG	AG/AA	*p*
N	84	60	n.s.	34	43	0.010
mHR	65.7 ± 12.5	64.5 ± 9.8	n.s.	74.9 ± 9.5	80.5 ± 12.2	0.033
LF/HF	2.6 ± 3.6	1.7 ± 1.7	n.s.	2.7 ± 2.2	3.4 ± 3.0	n.s.
RMSSD	63.0 ± 37.0	72.2 ± 53.1	n.s.	36.1 ± 29.5	34.7 ± 33.0	n.s.
Hc	0.82 ± 0.06	0.82 ± 0.07	n.s.	0.78 ± 0.08	0.77 ± 0.12	n.s.
QTc	0.414 ± 0.05	0.404 ± 0.04	n.s.	0.449 ± 0.05	0.453 ± 0.05	n.s.
QTvi	−1.30 ± 0.50	−1.41 ± 0.39	n.s.	−1.07 ± 0.55	−0.65 ± 0.76	0.008
	*KCNH2* rs3807374 T > G					
	Healthy Controls			Schizophrenia Patients		
MAF	G = 0.26 (75/288)			G = 0.34 (53/154)		
TT	0.58 (84/144)			0.44 (34/77)		
GT	0.31 (43/144)			0.43 (33/77)		
GG	0.11 (16/144)			0.13 (10/77)		
χ^2^	0.0417			0.0024		
CAF	TT	GT/GG	*p*	TT	GT/GG	*p*
N	85	59	n.s.	34	43	n.s.
mHR	65.7 ± 12.4	64.1 ± 9.8	n.s.	75.9 ± 10.5	79.7 ± 11.9	n.s.
LF/HF	2.7 ± 3.6	1.6 ± 1.4	n.s.	2.7 ± 2.1	3.4 ± 3.0	n.s.
RMSSD	64.5 ± 39.4	70.5 ± 51.2	n.s.	35.9 ± 29.7	34.8 ± 32.9	n.s.
Hc	0.82 ± 0.06	0.82 ± 0.07	n.s.	0.78 ± 0.07	0.77 ± 0.12	n.s.
QTc	0.414 ± 0.05	0.403 ± 0.04	n.s.	0.452 ± 0.05	0.451 ± 0.05	n.s.
QTvi	−1.32 ± 0.48	−1.40 ± 0.39	n.s.	−1.02 ± 0.62	−0.69 ± 0.74	n.s.
	*KCNH2* rs3807373 G > A					
	Healthy Controls			Schizophrenia Patients		
MAF	A = 0.25 (71/288)			A = 0.33 (51/154)		
GG	0.58 (84/144)			0.45 (35/77)		
AG	0.34 (49/144)			0.43 (33/77)		
AA	0.08 (11/144)			0.12 (9/77)		
^χ2^	0.0087			0.0012		
CAF	GG	AG/AA	*p*	GG	AG/AA	*p*
N	84	60	n.s.	35	42	0.024
mHR	65.7 ± 12.5	64.5 ± 9.8	n.s.	74.9 ± 9.4	80.6 ± 12.3	0.028
LF/HF	2.6 ± 3.6	1.7 ± 1.7	n.s.	2.7 ± 2.1	3.5 ± 3.1	n.s.
RMSSD	63.0 ± 37.0	72.2 ± 53.1	n.s.	36.1 ± 29.1	34.6 ± 33.4	n.s.
Hc	0.82 ± 0.06	0.82 ± 0.07	n.s.	0.78 ± 0.08	0.77 ± 0.12	n.s.
QTc	0.414 ± 0.05	0.404 ± 0.04	n.s.	0.448 ± 0.05	0.454 ± 0.05	n.s.
QTvi	−1.30 ± 0.50	−1.41 ± 0.39	n.s.	−1.05 ± 0.56	−0.66 ± 0.77	0.013
	*KCNH2* rs3800779 T > C					
	Healthy Controls			Schizophrenia Patients		
MAF	C = 0.30 (85/288)			C = 0.25 (39/154)		
TT	0.50 (72/144)			0.42 (32/77)		
CT	0.41 (59/144)			0.40 (31/77)		
CC	0.09 (13/144)			0.18 (14/77)		
χ^2^	0.0002			0.0228		
CAF	TT	CT/CC	*p*	TT	CT/CC	*p*
N	72	72	n.s.	32	45	0.018
mHR	66.6 ± 13.2	63.8 ± 9.3	n.s.	74.0 ± 9.3	80.9 ± 12.0	0.008
LF/HF	2.8 ± 3.8	1.7 ± 1.5	n.s.	2.9 ± 2.2	3.3 ± 3.0	n.s.
RMSSD	63.8 ± 39.9	69.9 ± 48.8	n.s.	36.3 ± 29.6	34.5 ± 32.8	n.s.
Hc	0.83 ± 0.06	0.81 ± 0.07	n.s.	0.78 ± 0.08	0.77 ± 0.11	n.s.
QTc	0.417 ± 0.05	0.403 ± 0.04	n.s.	0.446 ± 0.05	0.455 ± 0.05	n.s.
QTvi	−1.27 ± 0.52	−1.42 ± 0.39	n.s.	−1.06 ± 0.57	−0.68 ± 0.75	0.019
	*KCNH2* rs748693 A > G					
	Healthy Controls			Schizophrenia Patients		
MAF	G = 0.31 (88/288)			G = 0.40 (62/154)		
AA	0.49 (70/144)			0.38 (29/77)		
AG	0.42 (60/144)			0.44 (34/77)		
GG	0.09 (14/144)			0.18 (14/77)		
χ^2^	0.0000			0.0069		
CAF	AA	AG/GG	*p*	AA	AG/GG	*p*
N	70	74	n.s.	29	48	0.025
mHR	66.5 ± 13.3	63.9 ± 9.2	n.s.	74.7 ± 9.3	80.0 ± 12.1	0.049
LF/HF	2.8 ± 3.8	1.8 ± 1.6	n.s.	3.1 ± 2.2	3.1 ± 2.9	n.s.
RMSSD	64.5 ± 40.2	69.0 ± 48.4	n.s.	34.4 ± 29.5	35.8 ± 32.6	n.s.
Hc	0.83 ± 0.06	0.81 ± 0.07	n.s.	0.77 ± 0.08	0.78 ± 0.11	n.s.
QTc	0.417 ± 0.05	0.404 ± 0.04	n.s.	0.450 ± 0.05	0.452 ± 0.05	n.s.
QTvi	−1.26 ± 0.52	−1.42 ± 0.38	n.s.	−1.05 ± 0.57	−0.71 ± 0.75	0.042
	*KCNH2* rs1036145 C > T					
	Healthy Controls			Schizophrenia Patients		
MAF	T = 0.25 (73/288)			T = 0.39 (60/154)		
CC	0.51 (74/144)			0.39 (30/77)		
CT	0.40 (57/144)			0.44 (34/77)		
TT	0.09 (13/144)			0.17 (13/77)		
χ^2^	0.0008			0.0057		
CAF	CC	CT/TT	*p*	CC	CT/TT	*p*
N	74	70	n.s.	30	47	0.043
mHR	66.5 ± 13.3	63.9 ± 9.2	n.s.	75.0 ± 9.9	79.9 ± 11.9	n.s.
LF/HF	2.8 ± 3.8	1.8 ± 1.6	n.s.	3.2 ± 2.8	3.0 ± 2.7	n.s.
RMSSD	64.5 ± 40.2	69.0 ± 48.4	n.s.	35.8 ± 31.1	35.0 ± 31.7	n.s.
Hc	0.83 ± 0.06	0.81 ± 0.07	n.s.	0.78 ± 0.08	0.77 ± 0.11	n.s.
QTc	0.417 ± 0.05	0.404 ± 0.04	n.s.	0.450 ± 0.05	0.452 ± 0.05	n.s.
QTvi	−1.26 ± 0.52	−1.42 ± 0.38	n.s.	−1.04 ± 0.54	−0.71 ± 0.77	0.042

Distribution of allele frequencies. Other data expressed as mean (SD). *p*-values resulting from MANOVAs and follow-up ANOVAs. χ^2^ indicates the results from the Hardy–Weinberg equilibrium test. Abbrev.: Not significant (n.s.), Minor Allele Frequency (MAF), Mean Heart Rate (mHR), Root Mean Sum of Squared Distance (RMSSD), Heart Rate Low-Frequency/High-Frequency ratio (LF/HF), Compression entropy (Hc), Mean QT interval corrected for heart rate (QTc), QT variability index (QTvi).

## Data Availability

The datasets generated and/or analyzed during the current study are not publicly available but are available from the corresponding author on reasonable request.

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
