# Peer review of "Analysis of CACNA1C and KCNH2 Risk Variants on Cardiac Autonomic Function in Patients with Schizophrenia"

_genes, 2022, doi:10.3390/genes13112132_

Round 1

Reviewer 1 Report

Hi Dears;

1-What is the reason for choosing DSM-IV diagnostic criterion? 

(The DSM-IV is old criterion and DSM-IV TR and DSM-5  are newer).

2-  Please specify the age range of the patients participating in the research.

(Considering that age is an important intervening variable in Heart diseases).

3- Ethical point: Explain the reason for discontinuing the medication of Schizophrenic patients for 8 weeks?

( Discontinuing the medication of a Schizophrenic patient, can cause the  disorder to relapse and increase the risk of behavioral disorders).

4-Explain more about ,how to sampling?

Author Response

Please see the attachment. Thank you for your comments!

Reviewer 2 Report

I just have minor revision for authors:

1) Authors need to generate a summary table for all variants very clear to understand. 

Author Response

Please see the attachment. Thank you for your comment!

Round 2

Reviewer 1 Report

Thanks for the corrections.